# 6mA DNA Methylation on Genes in Plants Is Associated with Gene Complexity, Expression and Duplication

**DOI:** 10.3390/plants12101949

**Published:** 2023-05-10

**Authors:** Yue Zhang, Qian Zhang, Xingyu Yang, Xiaofeng Gu, Jinming Chen, Tao Shi

**Affiliations:** 1CAS Key Laboratory of Aquatic Botany and Watershed Ecology, Wuhan Botanical Garden, Chinese Academy of Sciences, Wuhan 430074, China; 2Center of Conservation Biology, Core Botanical Gardens, Chinese Academy of Sciences, Wuhan 430074, China; 3Biotechnology Research Institute, Chinese Academy of Agricultural Sciences, Beijing 100081, China; 4Wuhan Institute of Landscape Architecture, Wuhan 430081, China; 5Hubei Ecology Polytechnic College, Wuhan 430200, China

**Keywords:** *N*^6^-methyladenine, gene duplication, gene expression, *Nelumbo nucifera*

## Abstract

*N*^6^-methyladenine (6mA) DNA methylation has emerged as an important epigenetic modification in eukaryotes. Nevertheless, the evolution of the 6mA methylation of homologous genes after species and after gene duplications remains unclear in plants. To understand the evolution of 6mA methylation, we detected the genome-wide 6mA methylation patterns of four lotus plants (*Nelumbo nucifera*) from different geographic origins by nanopore sequencing and compared them to patterns in *Arabidopsis* and rice. Within lotus, the genomic distributions of 6mA sites are different from the widely studied 5mC methylation sites. Consistently, in lotus, *Arabidopsis* and rice, 6mA sites are enriched around transcriptional start sites, positively correlated with gene expression levels, and preferentially retained in highly and broadly expressed orthologs with longer gene lengths and more exons. Among different duplicate genes, 6mA methylation is significantly more enriched and conserved in whole-genome duplicates than in local duplicates. Overall, our study reveals the convergent patterns of 6mA methylation evolution based on both lineage and duplicate gene divergence, which underpin their potential role in gene regulatory evolution in plants.

## 1. Introduction

DNA methylation is a fundamental process of epigenetic regulation that can carry inheritable genetic information with some functional consequences beyond the four canonical DNA bases in plants [1,2,3]. The commonly studied 5-methylcytosine (5mC) is an epigenetic mark that is involved in many critical biological processes through gene regulation and has been extensively and widely studied in eukaryotes [2,3,4]. In contrast, *N*^6^-methyladenine (6mA), although being discovered at the same time as 5mC, has only received recent attention in eukaryotes, but it has been determined to be one of the most prevalent DNA modifications in prokaryotes [5,6]. 6mA together with *N*^4^-methyldeoxylcytosine are primarily used by prokaryotes in the restriction–modification system to protect the self-genome from foreign DNA invasion [5,7]. In prokaryotes, most 6mA sites are enriched in palindromic sequences and are involved in DNA replication, repair, and cell cycle regulation [8,9,10,11]. Given recent innovations in 6mA detection technologies, studies on the genome-wide distribution and function of 6mA modifications in plants started with model plant species and several crops, including the unicellular green algae *Chlamydomonas reinhardtii*, *Arabidopsis thaliana* (*Arabidopsis*), *Oryza sativa* (rice), and cotton [6,12,13,14,15]. 6mA is widely and evenly distributed across the genome of *Caenorhabditis elegans* but is enriched in transposable elements in the *Drosophila melanogaster* genome [16,17]. In plant genomes, conservation of 6mA enrichment around the transcription start site is correlated with active gene expression, such as in *C. reinhardtii*, *Arabidopsis*, and rice [6,12,18]. The development of methodologies for detecting modified bases from ONT (Oxford Nanopore Technologies) data offers an avenue for 6mA identification [19,20,21,22]. Moreover, similar to that of 5mC, the 6mA methylation level can vary in different tissues and respond to multiple abiotic stresses, as observed in *Arabidopsis* and rice [12,18].

DNA methylation plays a crucial role in gene regulation and an important role during species adaptation. Nevertheless, the evolution of methylation during lineage divergence has been primarily focused on 5mC. The divergence of 5mC methylation was suggested to be the initial substrate for selection and the maintenance of species boundaries during evolutionary divergence [23,24], and the variation in specific methylation sites in the promoter regulated the expression of the key genes that often contributed to speciation [25]. The 5mC methylation in the gene bodies of orthologs was mostly conserved between related species, such as *Brachypodium distachyon* and rice [26]. Nevertheless, the evolution of 6mA methylation on genes under plant species divergence remains to be addressed.

Gene duplication, including small-scale duplications and whole-genome duplications [27,28], increases gene dosage and diversifies gene functions [29,30]. Methylation patterns (primarily 5mC) can also evolve during paralogous divergence of different types of gene duplications to diversify gene expression and functions [31,32,33,34,35,36,37]. The extent of DNA methylation divergence within each duplicate pair was determined to be correlated with the evolutionary age of different types of duplicate genes [38], while the methylation levels in the gene body showed distinct distribution patterns among different types of duplicates [31,32]. The distinct methylation levels for duplicate partners were discovered to be associated with the expression differences in duplicates across different tissues [38,39,40]. However, we do not know whether the evolution of 6mA after gene duplication shows similar patterns as that of 5mC and the degree of conservation of 6mA in orthologous genes during plant speciation.

Sacred lotus (*Nelumbo nucifera* Gaertn., or lotus) is an early diverging eudicot plant that shows the most conserved genome architecture among eudicots and traces of only a single whole-genome duplication (WGD) [31,41], making it an ideal system to study gene fates after WGD. The recent chromosomal assembly of the *N. nucifera* (lotus) genome provides a framework for evolutionary genomic and epigenomic studies [31,42]. Although our previous study of the 5mC methylome showed distinct 5mC methylation patterns that are associated with gene expression [43], the cryptic 6mA distributions in lotus are still unclear. To address these questions regarding 6mA evolution in plants, we investigated the genome-wide 6mA variation and conservation in lotuses based on four newly assembled high-quality lotus genomes. More intriguingly, we revealed the convergence of 6mA evolution in lotus, *Arabidopsis* and rice genes during both orthologous and orthologous divergence.

## 2. Results

### 2.1. Genome Assemblies of Four Lotuses

To explore the 6mA distribution, diversity and evolution of different lotus genomes, we first performed genomic sequencing of four wild lotus plants with the Oxford Nanopore platform, yielding a total of 5.9 million single-molecule nanopore long reads with a data size of 183.32 Gb (Appendix A). Approximately 50× Illumina short reads for each lotus plant were generated for further hybrid assembly and polishing. Using the nanopore (long reads) and Illumina (short reads) data, we applied a hybrid assembly strategy, which produced genome assemblies for Indian lotus (421 contigs, N50 = 9.31 Mb), Australian lotus (471 contigs, N50 = 8.98 Mb), Russian lotus (1608 contigs, N50 = 2.36 Mb), and Thai lotus (1117 contigs, N50 = 3.87 Mb) (Table 1). Furthermore, the contigs for each lotus were anchored and ordered based on the reference genome “China Antique”, which produced final chromosomal assemblies for Indian lotus (764.6 Mb, GC = 38.86%), Australian lotus (769.67 Mb, GC = 38.88%), Russian lotus (790.18 Mb, GC = 38.89%), and Thai lotus (806.58 Mb, GC = 38.87%) (Table 1; Appendix A). The completeness of the genome assemblies was evaluated by plant conserved single-copy genes from BUSCO, which all showed BUSCO scores above 90% (Table 1).

Since the protein-coding genes of lotus var. “China Antique” have been fully annotated, we predicted the gene regions of these four lotus genomes by orthologous gene transfer. A final set of 34,966 protein-coding genes were annotated in Indian lotus, with 35,093 in Australian lotus, 37,129 in Russian lotus, and 36,314 in Thai lotus (Appendix A). Among these annotated genes in the four genomes, the average gene length was 1454.38 bp (SD = 1277). Almost 80% of these genes were identified as high-confidence genes, either with homology to other plants or support from RNA-seq data, suggesting the high credibility of our annotations. Furthermore, 56.03% of the assembled Indian lotus genome was annotated as repetitive elements, with 56.19% for Australian lotus, 55.74% for Russian lotus, and 56.36% for Thai lotus. In the four lotuses, long terminal repeat retrotransposons (LTRs) were the most abundant, followed by long interspersed nuclear elements (LINEs) (Appendix A).

### 2.2. Patterns of Genome-Wide Distribution of 6mA in Four Lotuses

According to the electrical signal change in the eukaryote model, the raw electric signals of Oxford Nanopore long-read sequencing data with mean 30.97× read coverage (Appendix A) were used to identify individual 6mA sites along with nanopolish in the four wild lotus genomes [44]. After methylation site calling and filtering, 3,698,642 (1.50% of all A sites), 3,759,921 (1.48%), 2,937,388 (1.19%), and 3,285,162 (1.27%) 6mA sites were identified in Indian, Russian, Australian, and Thai lotus, respectively (Figure 1A). The densities of 6mA (6mA/A) in the four wild lotus genomes were higher than those in *Arabidopsis* (0.04%) [12], rice (0.15–0.55%) [18], and soybeans (0.04%) [45] but lower than those in some early-diverging fungi, such as *Hesseltinella vesiculosa* (2.8%) [14]. Moreover, we detected a conserved motif, “WTAAK” (W = A/T, K = G/T), based on the 4 bp sequences upstream and downstream of the 6mA sites by using MEME-ChIP; this is the most significantly enriched motif in the four lotus genomes (Appendix A). In addition, we remapped the whole-genome bisulfite sequencing datasets of the four wild lotuses to their corresponding genomes [43] and identified genome-wide 5mC sites in three sequence context CG, CHG, and CHH (where H = A, T, or C), which are also present in high abundance in the four lotus genomes (Appendix A). The distribution analysis of 5mC sites near 6mA sites indicated that no correlation between them was detected, which is in line with reports in rice and *Chlamydomonas* [6,18] (Figure 1B). In contrast, we analyzed the percentage of 6mA sites in different 5mC contexts (see M&M). Our results suggested extremely low 6mA methylation levels in the three 5mC contexts, but even so, the adenines were more likely 6mA methylated in CGN contexts (where N = A, T, C, or G) than in CHG and CHH contexts (χ^2^ test, *p*-value < 0.01) (Appendix A).

By combining the genome annotation with 6mA sites, we determined that 6mA sites are more densely distributed in genic regions than in transposable elements (Appendix A). To further investigate the distribution of 6mA in the functional elements in lotus genomes, we divided lotus genomes into intergenic regions, promoters, and gene bodies, and the gene bodies were further broken down into exons and introns. Compared with the other wild lotuses, the Russian lotus had the highest percentage of 6mA located on gene bodies (41.55%), followed by the Australian lotus (39.89%) (Figure 1C). However, given that gene bodies occupy, on average, only 28.89% of the entire genome length for the four lotuses, this suggests that 6mA methylation occurs more frequently in gene body regions (χ^2^ test, *p*-value < 0.01). By combining the results for the gene body regions with 6mA sites, a total of 23,373 genes were determined to be 6mA-methylated in Indian lotus, with 18,300 in Australian lotus, 20,356 in Russian lotus, and 19,790 in Thai lotus (Figure 1C). In addition, we summarized the number of 6mA sites in each gene body region, and the most common 6mA-methylated genes contained fewer than five 6mA sites (Appendix A).

To reveal the features of adenine methylation specificity, we calculated the 6mA occupancy, representing the percentage of 6mA sites out of the total adenine sites, for each 50 bp window surrounding the gene start site (GSS) and gene end site (GES) in four lotus genomes (Appendix A). Intriguingly, an apparent pattern of 6mA distribution 2 kb upstream and downstream of the GSS was observed, while there was no specific pattern near the GES (Appendix A). The 6mA sites displayed a general trend of enrichment near the GSS, but a discontinuity between peaks upstream and downstream of the GSS resulted in a small bimodal distribution pattern. Furthermore, we analyzed the adenine frequency distribution around the 2 kb upstream and downstream sequences of the GSS (Appendix A). We observed significant degradation near the GSS, suggesting that the enrichment of 6mA sites around the GSS in genes is specific and not caused by an adenine bias. In addition, we analyzed the frequency distribution of 6mA sites in different transposable element (TE) families and repeat sequences. Briefly, the 6mA sites were enriched at the start site and end site of CMC-EnSpm, and the Copia and Gypsy families had higher methylation levels in functional regions. The LINE-L1 family had lower methylation levels in repeat regions, while no specific pattern was detected in other families (Appendix A). These distribution patterns were observed in all wild lotus genomes, suggesting that the 6mA distribution in transposable elements and repeat regions was conserved in lotus.

We discovered that over 52% of the annotated genes in the four wild lotus genomes were 6mA methylated (at least one 6mA site in gene body region). We compared the 6mA-methylated genes across the four wild lotus genomes using a Venn diagram, and a total of 7393 genes were commonly methylated, suggesting high conservation of 6mA methylation in lotus genes (Figure 1D). The Indian lotus had the most specific 6mA-methylated genes, while the Australian lotus had the least. To provide insights into the function of 6mA-methylated genes in lotus, the Gene Ontology (GO) enrichment results revealed that the 6mA genes were involved in multiple biological processes, particularly in chromosome organization and DNA damage repair, suggesting that the 6mA-methylated genes likely play a vital role in the regulation of genes involved in genetic material replication (Appendix A).

### 2.3. The Relationships between 6mA and Gene Expression in Lotus

To investigate the relationships between 6mA and gene expression, we carried out transcriptome sequencing on four lotus plants with two biological replicates each. The FPKM box plot showed that genes with 6mA sites (6mA genes) were expressed at significantly higher levels than those without 6mA sites (non-6mA genes) in lotus genomes (*t*-test, *p*-value < 10^−5^) (Figure 2A–D; Appendix A). We determined that genes with high 6mA methylation levels (over 100 6mA sites) had higher expression levels than genes with intermediate 6mA methylation levels (1~99 6mA sites) and non-6mA genes (Appendix A). We further performed GO enrichment analysis of genes with high 6mA methylation levels in four lotuses, and results indicated their function involved in main biological metabolic processes, notably including an intriguing GO term “regulation of gene expression in epigenetics” (Appendix A). Moreover, based on the screening criteria for gene expression in rice, we also divided the gene expression levels into high expression (FPKM ≥ 1) and low expression (FPKM < 1). We determined that significantly more 6mA-methylated genes were highly expressed in all lotuses (chi-square test, all *p*-values < 0.01) (Appendix A). In addition, genes without 6mA sites were expressed at a low level in all lotus genomes (Appendix A). Consistently, the 6mA sites occurred more frequently in the gene bodies of highly expressed genes than in those of genes with low expression (Figure 2E–H). In contrast, gene promoters with 6mA and without 6mA showed no significant difference in their expression levels (*t*-test, *p*-value > 10^−5^) (Appendix A). However, the 6mA methylation levels in GSS upstream and GES downstream regions of genes with low expression were higher than those of highly expressed genes (Figure 2E–H). Therefore, our results suggested that 6mA modifications on gene bodies are associated with active expression in lotus.

### 2.4. Higher Gene Structural Complexity, Expression Level, and Expression Breadth for 6mA-Methylated Genes in Plants

To explore whether there are shared features in the gene structure and expression of 6mA-methylated genes in plants, we further included the data for well-annotated 6mA-methylated sites in the model plants *Arabidopsis* [12] and rice [18], which have diverged for more than 100 million years. All genes in lotus, *Arabidopsis*, and rice were split into two groups: 6mA genes (carrying at least one 6mA site) and non6mA genes (carrying no 6mA sites). For lotus, 6mA genes were designated as those with 6mA methylation in any of the four lotuses considered. Moreover, genes were divided into four groups from small to large according to their quartiles for gene length, exon number, CDS length, gene expression (FPKM), and tissue specificity (tau index). Through pairwise between-group comparisons, genes in the group with a longer gene length (>3000 bp) (Figure 3A–C) and more exons (>7) (Figure 3D–F) had a significantly (chi-square test, *p* < 0.01) higher percentage of 6mA modifications across all three species, indicating that longer and multiple-exon genes tend to have a higher probability of possessing 6mA sites. We also determined that genes translating longer proteins (CDS length > 1500 bp) contained significantly (chi-square test, *p* < 0.01) more 6mA-modified genes than genes with shorter CDSs in lotus and *Arabidopsis* (Figure 3G–H), whereas genes whose CDSs ranged from 1000 bp to 1500 bp had the most 6mA-modified genes in rice (Figure 3I), suggesting the divergence of 6mA modification between monocots and dicots in the coding region of genes. Other than gene structure-based features, in line with previous results indicating that 6mA methylation was related to the high expression of genes [6,12,18], our results showed that genes expressed at a low level (FPKM < 5) have significantly (chi-square test, *p* < 0.01) fewer 6mA-modified genes in all three species (Appendix A). In addition, the tau index was used to assess the tissue specificity of genes in the three species (see M&M). Our results suggested that genes with a lower tau index (<0.25) or broader tissue expression had a significantly (chi-square test, *p* < 0.01) higher percentage of 6mA-modified genes than genes with higher tissue specificity (Figure 3J–L). Furthermore, we randomly selected genes in lotus chromosome1 and chromosome5 to perform the above analyses as control datasets for expectation under the null hypothesis, and the results of sampled genes were similar to those of all genes (Appendix A). In addition, we should also note that the gene features of 6mA-modified genes in each of the four lotus genomes showed a consistent trend across the above analyses, which used the combined 6mA-modified genes in all lotus genomes (Appendix A). These results suggested that the patterns of 6mA methylation are stable.

To further explore the 6mA modification of genes from gene families of different sizes in plants, we focused on the orthologous groups (OGs) present in all three species. According to the number of gene copies in each OG, the genes in lotus, *Arabidopsis*, and rice were divided into four groups according to the OG size (copy number). Interestingly, the smaller OGs (with fewer than two gene copies) had a significantly (chi-square test, *p* < 0.01) higher overall percentage of 6mA-modified genes (Appendix A), indicating that 6mA modifications were more likely to occur in genes undergoing fewer duplication events. Furthermore, for each of the three species, we focused on the OGs that included both the 6mA-methylated genes and non-6mA-methylated genes and compared the gene features between these two gene groups. Compared to genes without 6mA modification, the 6mA-modified genes had longer lengths and more exons in all three species (Mann–Whitney *U* tests, *p* < 0.01) (Appendix A). Moreover, 6mA-modified genes had significantly (Mann–Whitney *U* test, *p* < 0.01) longer CDSs and lower tissue-specific expression (tau index) than non-6mA-modified genes in lotus and *Arabidopsis*, whereas no difference in CDS length and tissue specificity was detected between 6mA-modified and non-6mA-modified genes in rice (Appendix A). Notably, the 6mA-modified genes had significantly (Mann–Whitney *U* test, *p* < 0.01) higher expression levels than the non-6mA-modified genes in all three species (Appendix A). These results suggested that 6mA modification generally does not tend to be maintained after gene duplication for genes with shorter lengths, fewer exons or shorter genes with lower expression.

### 2.5. Evolution of 6mA Modification in Duplicated Genes from Different Origins

Our previous study suggested that 5mC methylation levels and patterns vary substantially among different types of duplicate genes in the lotus genome and are associated with corresponding gene expression behaviors [31]. Herein, we investigated both the conservation and divergence of 6mA modification associated with different types of duplications in lotus, *Arabidopsis*, and rice, all of which experienced whole-genome duplications (WGDs) [31,46]. First, different types of duplicated genes were identified in the three species (see M&M), and for each species, the genes with no homologous genes in other plant species in the PLAZA database [47] were designated orphan genes. We calculated the percentage of 6mA-modified genes in different gene groups and determined that singleton (single-copy) genes had the highest percentage of 6mA-modified genes in both lotus and rice, whereas dispersed genes in *Arabidopsis* had the most 6mA genes. In contrast, the orphan genes had the lowest percentage of 6mA-modified genes in all three species (Figure 4A–C). Moreover, the local duplicates (tandem and proximal) had a lower percentage of 6mA-modified genes than WGD/segmental duplicates in lotus, *Arabidopsis*, and rice (Figure 4A–C). We observed that the percentage of 6mA-modified genes in different types of duplications varied among the three species (Figure 4A–C), which might be due to their different duplication events.

We further studied how 6mA modification is maintained between the closest duplicate gene pairs (hereafter named paralogs). We determined that the 6mA modification in locally duplicated gene pairs was significantly (chi-square test, *p* < 0.01) more likely to be changed (showing presence and absence between copies) than that in either WGD or dispersed duplicated gene pairs in the three species (Figure 4D–F, Appendix A). Moreover, significantly more WGD duplicate pairs than dispersed duplicated gene pairs maintained the 6mA modification (in both copies) in lotus (Figure 4D). However, in *Arabidopsis* and rice, this situation was reversed, in which more dispersed duplicate pairs than WGD duplicate pairs maintained the 6mA modification (in both copies) (Figure 4E,F) (chi-square test, *p* < 0.05 in rice). Such a difference among species is likely because the ages and episodes of WGDs are different for the three species: lotus underwent only one WGD, *Arabidopsis* had three WGDs, and rice had four WGDs [31,46].

### 2.6. Evolution of 6mA Methylation on Orthologous Genes after Long-Term Species Divergence

To understand how 6mA methylation changes or is maintained in orthologous genes among distantly related plant taxa, we performed pairwise analysis of 6mA methylation between orthologous genes among lotus, *Arabidopsis* and rice. Considering that not all orthologs have a one-to-one relationship, we chose only the orthologous gene pairs that was also the best hit in the BLAST search in the following comparisons. Intriguingly, in both comparisons of lotus vs. *Arabidopsis* and lotus vs. rice, significantly more orthologous genes in *Arabidopsis* (or rice) corresponding to lotus genes with 6mA modifications were 6mA modificated than those lotus genes without a 6mA modification (chi-square test, *p* < 0.01) (Figure 5A,B, Appendix A), i.e., lotus genes with 6mA tended to maintain their 6mA methylation status even after >100 million years of evolution. Further GO functional enrichment analysis suggested that the orthologous genes with 6mA modifications maintained in all three species are linked to plant tissue development processes (Appendix A). We also detected a significantly higher *Ka*/*Ks* ratio in the orthologous gene pairs with a 6mA change (presence/absence) than in those pairs with 6mA maintenance for both the lotus vs. *Arabidopsis* and lotus vs. rice comparisons (Mann–Whitney *U* test, *p* < 0.01) (Figure 5C,D, Appendix A), suggesting that 6mA-modified genes are under higher purifying selection or constraint during species divergence.

## 3. Discussion

Although 5mC and 6mA methylation were both detected a long time ago [48], 6mA methylation has attracted more attention from researchers in recent years given the technological innovations surrounding 6mA detection. Recent single-molecule sequencing platforms with long-read capabilities have revolutionized both genome sequencing and DNA methylation identification [49,50] and have been applied to investigate 6mA modifications in single nucleotide sites in model plants, microorganisms, and nonplant eukaryotes [6,12,18,19,51,52]. In this study, we de novo assembled lotus genomes from different geographic origins based on nanopore long reads, and in parallel, we detected 6mA signals based on nanopore sequencing [50,53,54]. Our results revealed a robust and genome-wide profile of 6mA sites in the four lotus genomes. The evolution of 5mC modifications during lineage divergence and adaptation has been studied extensively given its vast and common distribution and advances in 5mC detection technologies [55,56,57,58,59,60,61,62]. The 6mA distribution uncovered here in lotus broadens our current understanding of the roles of epigenetic modifications in the potential effects on gene regulation.

The genomic distribution patterns of 6mA sites in eukaryotic genomes are diverse, especially those discovered in recent animal studies [15,16,17,63]. However, previous studies in *Arabidopsis* and rice suggested that the 6mA distribution and consensus motifs are conserved [12,18]. Our results showed that the 6mA sites are enriched in exons and exhibit a small bimodal enrichment tendency around the GSS in the four lotus genomes, consistent with the results in *Chlamydomonas* and *Arabidopsis* [6,12]. Noticeably, the consensus sequence motifs for 6mA are highly conserved between the four wild lotuses, but most motifs in lotus are unique compared to those in *Arabidopsis* and rice [12,18]. Therefore, more patterns might be revealed as more plant genomes are sequenced using these high-throughput technologies. In addition, the effect of 6mA modification of DNA on genes involved in different biological processes is one of the central topics for 6mA studies. $N^{6}$-adenine methylation has been shown to regulate gene transcription by modifying transcription factor binding or altering chromatin structure in eukaryotes [49]. In barley, the expression level of 6mA-modified reporter plasmids was increased, whereas 5mC had no similar effect on transcription efficiency under a transient expression system [64]. Importantly, recent studies suggested that the enrichment of 6mA modifications around the GSS region positively correlates with gene expression levels in plant genomes, while 5mC modifications showed no specific enrichment pattern around the GSS region and did not colocalize with 6mA modifications [6,12,18]. In this study, we also tested the correlation between the gene expression level and its 6mA sites, and we determined that the 6mA sites in gene bodies are significantly positively correlated with gene expression for all genes in general, further indicating an association between 6mA modification intensity and gene expression level across the genome in plants. Yet, further molecular experiments by manipulating 6mA modifications are needed to verify the impact of 6mA modifications on gene expression level. In this study, we detected the common structural and expression level features of genes that have a 6mA modification in lotus, *Arabidopsis*, and rice; these genes show a longer gene length, more exons, and higher and broader expression. Long proteins often contain more functional and regulatory domains, while short proteins have related limited functionalities [65,66].

Previous studies on the DNA methylation of duplicated genes have mainly focused on the relationship between the divergence of 5mC methylation and gene expression levels in animal and plant species [39,67]. Here, we revealed that a higher level of 6mA is maintained in small orthologous groups (OGs) and that a stronger divergence of 6mA modification occurs for larger OGs in all three species. The divergence of DNA methylation between duplicated gene pairs facilitates the expression differences between them, thus playing a role in duplicate maintenance through subfunctionalization and neofunctionalization [34,36,38,40]. Given that 5mC methylation was distinct among different types of duplicate genes in our previous study [31], we also explored the evolution of 6mA methylation in different duplicates in lotus, *Arabidopsis*, and rice. We observed that the duplicated gene pairs produced by WGD and dispersed duplication are more easily maintained or generate novel 6mA modifications than genes from local duplications, likely due to the constraint of gene dosage balance [68]. In addition to duplicate genes, our ortholog analysis between the three flowering plants revealed high conservation of 6mA modifications in their closest orthologous gene pairs and purifying selection maintaining the 6mA modifications between orthologous genes during species divergence. Most orthologous genes with common 6mA modifications in all investigated species are highly expressed and involved in many tissue development processes in plants. In bacteria, 6mA-modified genes play an important role in the regulation of bacterial DNA replication and repair, transposition, and nucleoid segregation [69]. This result suggests that 6mA tends to preferentially modify different functional genes of plants in comparison to bacteria. However, the identification of more 6mA modifications in different plant lineages via Nanopore or PacBio sequencing will be necessary to understand the broader role of 6mA in plants.

## 4. Conclusions

Herein, we globally presented single-nucleotide resolution of 6mA modifications from Nanopore sequencing signals in four *Nelumbo* genomes and revealed the genomic distribution and evolutionary patterns of 6mA modifications across species and duplicate genes. Our findings revealed consistently local and global patterns of 6mA levels in four lotuses that are associated with corresponding expression rewiring, which highlights the positive role of 6mA modification around the GSS that is correlated to gene expression. These patterns of 6mA modification are preferentially retained in highly and broadly expressed genes with long lengths among distantly related plants. Intriguingly, 6mA modifications are more likely to be retained in WGD than locally duplicated genes and during the long-term evolution of plant species.

## 5. Materials and Methods

### 5.1. Plant Materials, Library Construction, and Genome Sequencing

Seeds from four wild *Nelumbo nucifera* plants were collected from Russia (132°24′ E, 42°51′ N), India (74°42′ E, 14°58′ N), Thailand (99°52′ E, 13°07′ N) and Australia (146°43′ E, 19°19′ S) and cultivated at the Wuhan Botanical Garden, Chinese Academy of Sciences (114°30′ E, 30°60′ N), China. Total genomic DNA was isolated from fresh leaves of *N. nucifera* using the DNeasy Plant Kit. A total of 10 µg of high molecular weight DNA was processed for constructing the library for the MinION flow cell according to ONT’s instructions. For each DNA library, sequencing was performed in Oxford Nanopore Technologies PromethION with MinKNOW (v1.4.2) software for approximately 48 h to obtain ~50× raw dataset. After filtering the sequences with high-quality scores, the resulting FAST5 files were converted to FASTQ files with the Albacore base caller (v3.0.1). Additionally, we generated ~50× Illumina short reads for assembly and polishing during hybrid genome assembly. Briefly, the total DNA of each sample was used to construct a library with an insert size of 450 bp, which was sequenced on the Illumina HiSeq 2000 platform according to a standard protocol.

### 5.2. Genome Assembly and Annotation

After quality control, the clean Illumina data were de novo assembled into contigs by using SparseAssembler with -k 95 -g 15 -TrimN 20 [70]. Then, the assembled contigs combined with the MinION (v1.4.2) long reads were hybrid-assembled into scaffolds using DBG2OLC with the parameters AdaptiveTh 0.05 [71]. To polish the scaffolds, we used the Pilon (v1.22) algorithm with default parameters to correct assembly errors with ~50× Illumina short reads for four different lotus genomes. The polished scaffolds were ordered and oriented into pseudochromosomes guided by their alignment to the “China Antique” reference genome [72] by using the nucmer (-g 90) and mScaffolder (-ul n) tools from MUMmer [73]. We finally assessed the completeness using the plant Benchmarking Universal Single-Copy Orthologs (BUSCO) dataset.

We first performed high-quality gene annotation filtering based on the “China Antique” reference [43] with the complete ORF and amino acid number >30 criteria. According to the orthologous relationship between the wild species and “China Antique”, the reference gene sequences were aligned to the wild genome using blat with -maxIntron 100,000 -minIdentity 90. The alignment results were excluded if the coverage was less than 90%. We performed the same strategy of genome assembly and gene annotation on all four wild genomes. Additionally, the high-quality gene sequences were mapped to the PLAZA [47] dataset by using Blastn with an *e*-value < 1 × 10^−6^, and only the best aligned sequence or the one with expression level evidence (from http://nelumbo.biocloud.net) was regarded as the high-confidence gene. Repeats and transposable elements in each genome were detected by de novo-based and homology-based strategies using RepeatModeler (v2.0) and RepeatMasker (v4.1.0) based on previously identified lotus repeats [72].

### 5.3. 6mA Modification Detection and Filtering

Oxford Nanopore sequencing detects 6mA DNA modifications at single-nucleotide resolution by comparing raw electric signals of methylated DNA copies with signals of the same unmethylated DNA copies. In brief, the clean nanopore reads from each wild species were first remapped to the corresponding genome assembly by minimap2 [74]. Then, we created the index file that links read ids with their signal-level data in the FAST5 files and detected methylated bases by using nanopolish with parameter eventalign (https://github.com/jts/nanopolish.git (accessed on 17 March 2022)). We used mCaller (https://github.com/al-mcintyre/mCaller (accessed on 17 March 2022)), a Python program that calls 6mA sites from nanopore signal data, to detect 6mA sites with a minimum coverage of 10×. The four assembled lotus genomes and 6mA locus information were deposited in Figshare (https://doi.org/10.6084/m9.figshare.13191506 (accessed on 17 March 2022)). The 6mA site data for *Arabidopsis* (*Col*) wild-type [12] and rice [18] were downloaded from previous studies, and the genes with 6mA modification in different ecotypes of rice were combined to comprehensively represent the 6mA-methylated genes in rice.

### 5.4. Whole-Genome 5mC Analysis

The whole-genome bisulfite sequencing datasets of lotus samples from our previous study were downloaded from NCBI with accession number PRJNA552416 [43]. The clean reads of each wild species were mapped to the corresponding assembly genome using bowtie2 [75], as implemented in Bismark [76]. Then, duplicates were removed, and DNA methylation calls of CG, CHG, and CHH (where H = A, T, or C) were extracted by using “deduplicate_bismark” and “bismark_methylation_extraxtor” (--comprehensive). After DNA methylation calling by Bismark, only 5mC sites covered by at least five reads were retained for further analysis. To explore the different 6mA methylation levels in the three 5mC contexts CGN, CHG, and CHH (where N = A, T, C, or G), we calculated the ratio of 6mA sites in these three contexts, of which we only counted one if two adenines were 6mA methylated in one CHH site, i.e., CA^m^A^m^.

### 5.5. RNA-seq and Expression Analysis

For each sample, total RNA was extracted using the RNAprep Pure Plant Kit. After quality checking by 1% agarose gels, the RNA concentration and integrity were examined. A total of 3 µg of eligible RNA from each sample was used for constructing the Illumina sequencing library. The library was then sequenced on an Illumina HiSeq 2000 platform, and 150 bp paired-end reads were generated. After quality control, the clean reads were mapped to the assembled reference genome using HISAT2 [77], and the FPKMs of the genes were calculated by StringTie under gene annotation with default parameters [78].

To assess the tissue specificity of a gene at the expression level, the gene expression matrices of multiple tissues were constructed for lotus, *Arabidopsis*, and rice based on their corresponding genome databases (http://nelumbo.biocloud.net (accessed on 17 March 2022) and http://expression.ic4r.org (accessed on 17 March 2022)) or multiple-tissue RNA-seq samples [72,79,80]. Herein, we used the tau index to measure the tissue-specific expression level of a gene across multiple tissues [81]:$$\mathrm{tau}=\frac{\sum_{i=1}^{n}\left(1-{\hat{x}}_{i}\right)}{n-1};\;{\hat{x}}_{i}=\frac{x_{i}}{{max}_{1\leq i\leq n}(x_{i})},$$
where x_i_ = log (FPKM of gene x in tissue i) and n = the number of tissues.

### 5.6. Classification of Different Types of Duplicated Genes and Identification of Orthologous Genes

To identify the duplicated genes from different origins, we divided the genes from each of the three studied species into five groups based on the MCScanX results: singletons, WGD/segmental duplications, dispersed duplications, proximal duplications, and tandem duplications. Moreover, the gene sequences of lotus, *Arabidopsis*, and rice were mapped to the PLAZA 4.0 database using BlastN with an *e*-value < 1 × 10^−6^after excluding duplicates, and the genes that showed no mapped results (or no orthologous genes) were further defined as orphan genes, most of which were transient and lineage-specific [82].

The orthologs among *Arabidopsis*, rice, and lotus were identified using OrthoMCL with an *e*-value < 1 × 10^−15^ and an inflation parameter of 2.0. Only orthologous groups (OGs) that contained genes from all three species were retained for further analyses. To identify the closest orthologous gene pairs, the protein sequences of lotus were mapped to *Arabidopsis* and rice using BlastP with a minimum identity > 0.9 and *e*-value < 1 × 10^−6^. Only the orthologous gene pairs that contained the best hit from the mapped results and that were detected in the same OG were considered the closest orthologous gene pairs. In addition, the divergence parameters (including *Ks* (synonymous substitution rate), *Ka* (nonsynonymous substitution rate), and *Ka/Ks*) of each of these orthologous gene pairs were calculated using KaKs Calculator [83].

### 5.7. GO Enrichment Analysis 

To annotate the functions of all genes, KOBAS2.0 [84] was used to map all gene sequences to the Gene Ontology (GO) database, and TBtools (https://github.com/CJ-Chen/TBtools (accessed on 17 March 2022)) was used to obtain the significantly enriched GO terms for each gene set.

## Figures and Tables

**Figure 1 plants-12-01949-f001:**
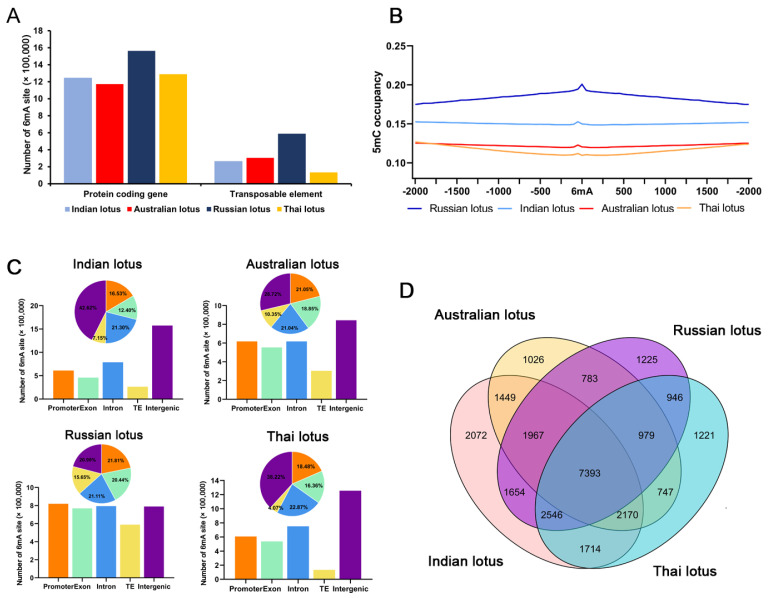
Distribution patterns of 6mA methylation in four wild *N. nucifera* (lotus) genomes. (**A**) Percentage of 6mA sites in all adenine in the four genomes. (**B**) Distribution of 5mC sites around the 6mA sites in four lotus genomes. The 5mC occupancy represents the 5mC sites out of the total cytosine sites in 50 bp sliding windows ±2 kb of the 6mA sites. (**C**) Distribution of 6mA sites in exons, introns, promoters (−2 kb of the gene start site (GSS)), and intergenic regions in four lotus genomes. (**D**) Venn diagrams showing overlaps among the genes with 6mA modifications in four lotus genomes.

**Figure 2 plants-12-01949-f002:**
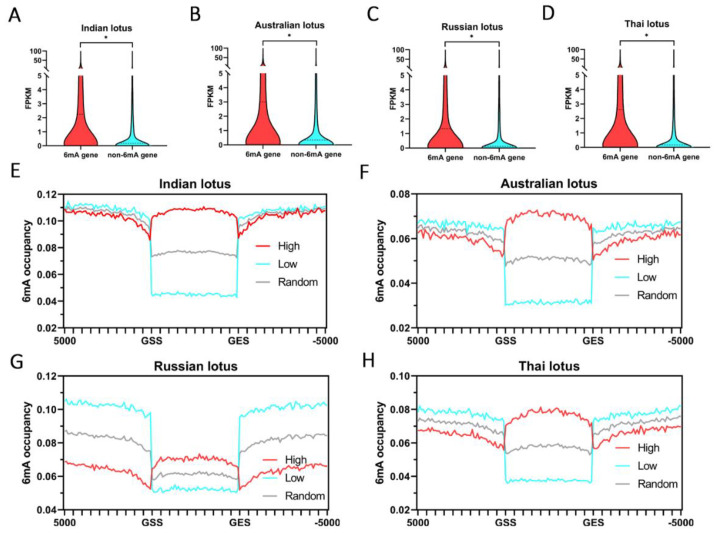
6mA methylation and gene expression levels in lotus. (**A**–**D**) Box plot comparing expression levels (FPKMs) between genes with and without 6mA sites in four lotus genomes. The *p* values are from two-tailed unpaired Student’s *t*-tests, and * means *p* values < 10^−5^. (**E**–**H**) Distribution of 6mA sites in genes with high expression level (FPKM > 1), genes with low expression level (FPKM < 1), and 10,000 random selected genes with both high- and low-expressed genes as control background level in four lotus genomes. The 6mA occupancy represents the 6mA sites out of the total adenine sites in 100 bp sliding windows within −5 kb of the GSS and +5 kb of the GES.

**Figure 3 plants-12-01949-f003:**
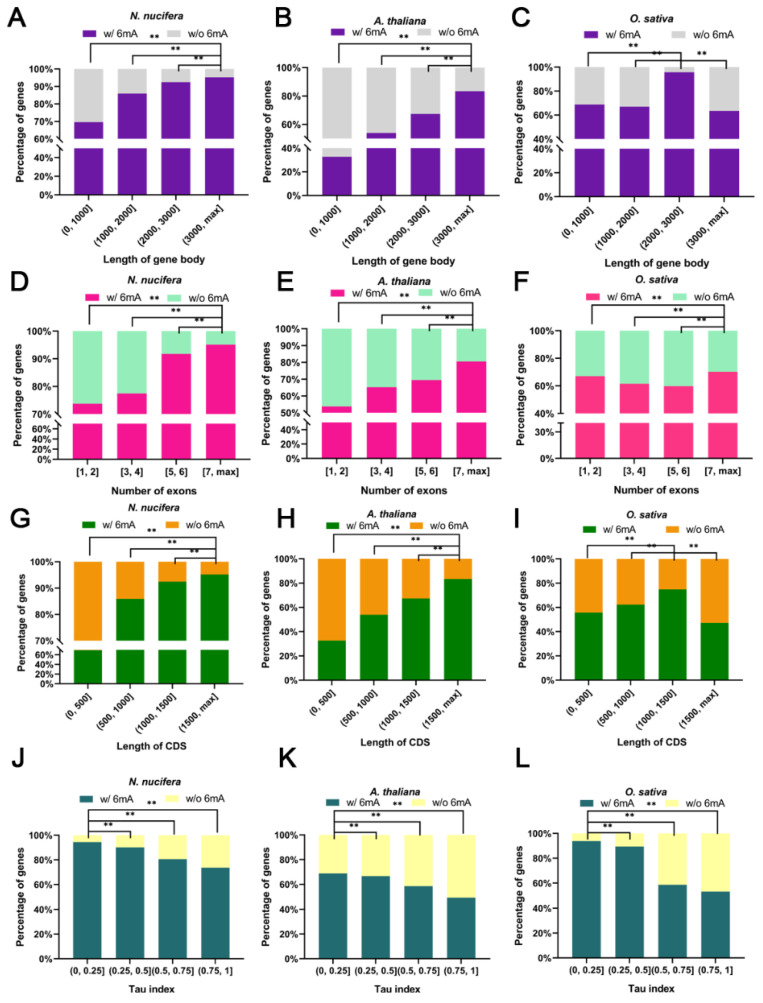
Contrast features of genes with and without 6mA-methylation in lotus, *Arabidopsis*, and rice. Comparative analysis of genes with 6mA modification or without 6mA modification based on different types of gene features, including gene length (**A**–**C**), exon number (**D**–**F**), CDS length (**G**–**I**), and tissue specificity (tau index) (**J**–**L**). The genes of each of the three species were divided into four groups from small to large according to their quartiles. The differences between different gene groups were tested by chi-square test, ** was *p*-value < 0.01.

**Figure 4 plants-12-01949-f004:**
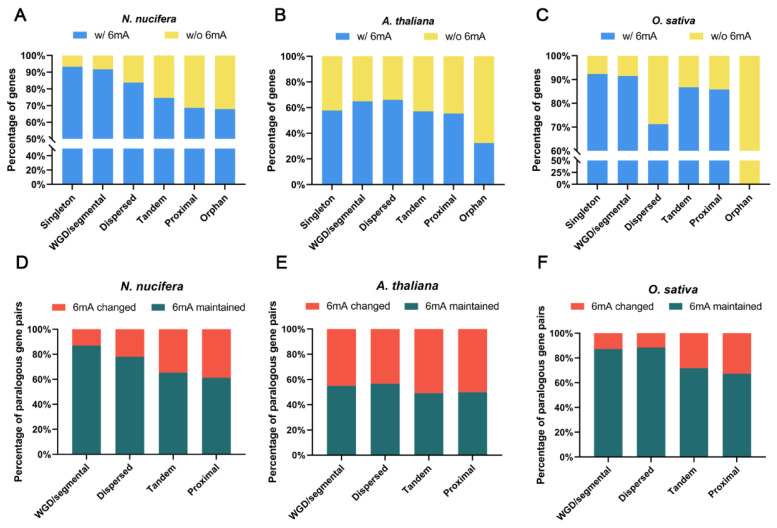
The evolution of 6mA methylation associated with different gene duplications in lotus, *Arabidopsis*, and rice. (**A**–**C**) The percentage of genes with or without 6mA methylation in gene groups of different duplication types in the three species. Singleton: genes without any homolog within the genome; orphan: genes without a homolog in any other species. (**D**–**F**) The duplicated gene pairs (the query gene and its closest paralogous gene) from different duplication types exhibit distinct proportions of gene pairs with the 6mA modification change and with the 6mA maintenance after duplication for the three species.

**Figure 5 plants-12-01949-f005:**
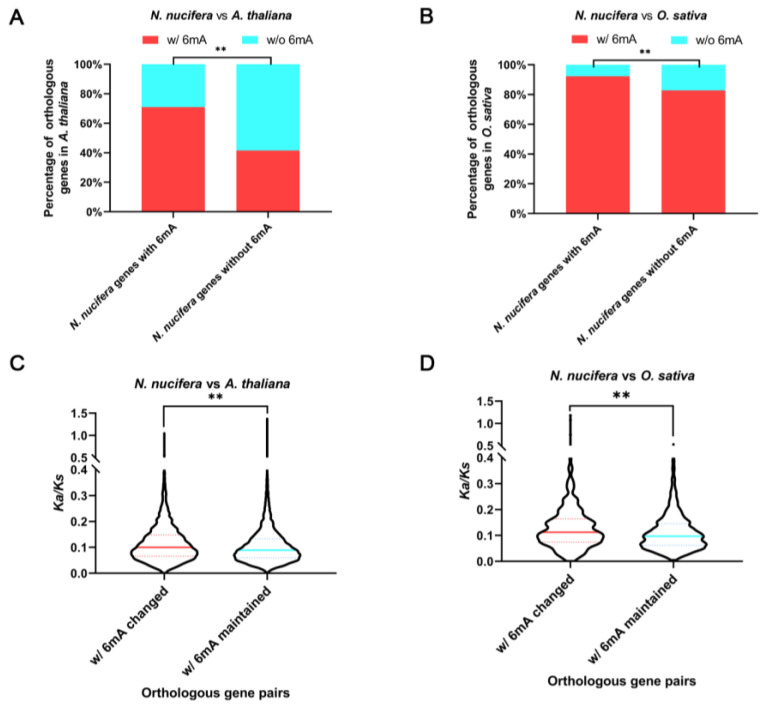
Maintenance of 6mA modifications between orthologous genes (lotus versus *Arabidopsis*, and lotus versus rice). (**A**,**B**) The orthologous genes of lotus 6mA genes in *Arabidopsis* (**A**) and rice (**B**) exhibit significantly higher proportions of 6mA modification than the orthologous genes of lotus genes without any 6mA. The significance was tested by chi-square test, and ** means *p* < 0.01. (**C**,D) The contrast *Ka/Ks* ratios between the orthologous gene pairs with 6mA changes and 6mA maintenance are shown in the violin plots for both lotus vs. *Arabidopsis* (**C**) and lotus vs. rice comparisons (**D**). The significance was tested by Mann–Whitney *U* test, and ** means *p* < 0.01.

**Table 1 plants-12-01949-t001:** Summary of the assembly and annotations for the Indian lotus, Australian lotus, Russian lotus, and Thai lotus genomes. BUSCO: Benchmarking Universal Single-Copy Orthologs.

Feature	Indian Lotus	Australian Lotus	Russian Lotus	Thai Lotus
Genome size (bp)	801,742,539	807,059,836	828,573,050	845,764,567
Contig number	421	471	1608	1117
Longest contig (bp)	40,391,228	30,603,528	14,404,078	23,450,770
Average contig size (bp)	1,903,041	1,708,757	513,269	756,434
N50	9,768,644	9,418,673	2,477,834	4,066,382
GC (%)	38.86	38.88	38.89	38.87
BUSCO (%)	91.8	91.6	91.1	90.9
Gene number	34,966	35,093	37,129	36,314
Exon number	149,316	151,500	160,545	166,936
Intron number	114,350	116,407	123,416	130,622
Average CDS length (bp)	1457.19	1458.5	1458.81	1443.17
High-confidence gene (%)	80.15	80.20	79.79	79.93
Repetitive elements (%)	56.03	56.19	55.74	56.36

## Data Availability

The assembled genomes of the four wild lotus samples used in this study and the raw nanopore datasets with electric signals have been deposited into the CNGB Sequence Archive (CNSA) [85] of the China National GeneBank DataBase (CNGBdb) with accession numbers CNP0001849. Sequence data from this article can also be found in the Sequence Read Archive (SRA) under accession numbers PRJNA633299, PRJNA633737, PRJNA633708, and PRJNA674489.

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
