# Peer review of "6mA DNA Methylation on Genes in Plants Is Associated with Gene Complexity, Expression and Duplication"

_plants, 2023, doi:10.3390/plants12101949_

Round 1

Reviewer 1 Report

Zhang and collaborators provide an interesting study on the N6 methylation in plant genomes, with the main study on four different types of Lotus plants with different recorded evolutionary history. As such the manuscript is interesting as well as some of the posible repercussions that this epigenetic tag in the genomes may be maintain during evolution.  Although most of the data is well presented figure 5a and 5b are not clear as to why if the colors represent with 6mA or without 6mA what are the x axis indicating as it seems to indicate the same, but then its not clear what are they stating, also the description of these results its not clear. Also it would be interesting to provide a simplistic model of the process from the genes that conserve this type of modification. Even if it is very general it may provide target areas for further research

Author Response

Responses:

1.Although most of the data is well presented figure 5a and 5b are not clear as to why if the colors represent with 6mA or without 6mA what are the x axis indicating as it seems to indicate the same, but then its not clear what are they stating, also the description of these results its not clear.

Response: We revised the x-axis in Figures 5a and 5b to show our results more clearly. The x-axis now shows “N. nucifera genes with/without 6mA”, and the colors represent those genes’ orthologs in Arabidopsis (or rice) with 6mA or without 6mA. Also, we revised the sentence in lines 421-425:

Lines 421-425: Intriguingly, in both comparisons of lotus vs. Arabidopsis and lotus vs. rice, significantly more orthologous genes in Arabidopsis (or rice) corresponding to lotus genes with 6mA modifications were 6mA modificated than those lotus genes without a 6mA modification (chi-square test, p< 0.01) (Fig. 5A-B, Supplemental Fig.S20A), i.e., lotus genes with 6mA tended to maintain their 6mA methylation status even after >100 million years of evolution.

2. Also it would be interesting to provide a simplistic model of the process from the genes that conserve this type of modification.

Response: We added a model diagram in Supplementary Figure S20A according to your comment. This model diagram shows how we analyze the orthologous genes with/without 6mA modification, making our results easier to understand.

Reviewer 2 Report

 In this manuscript from Y. Zhang et al, the authors analyze the genome-wide DNA 6mA methylation pattern in four wild lotus plants from India, Russia, Australia and Thailand. They found that 6mA sites were enriched at the transcriptional start sites, positively correlated with gene expression level, and tend to retain in highly and broadly expressed gene orthologs with longer length and more exons in lotus. Through comparing with 6mA patterns in Arabidopsis and Rice, the authors discovered that 6mA methylation was significantly more enriched and conserved in whole-genome duplicates than in local duplicates.

       Overall, this study is interesting and of interest for the Plants readership. The introduction section provides sufficient background and includes the most relevant references. The research design appropriately. The methods were properly described. Lots of data has been analyzed. This manuscript was well written and easy to follow. The following are some minor concerns that need to be modified:

1.       Line 45, What is ‘ONT data’? please explain.

2.       Line 45-46, the format of the references in the parenthesis is not consistent with others. Please keep them consistent. Besides, these references could not be found in the Reference section.

3.       Line 226, the Supplemental Fig. S4 is the consensus motif, which could not be cited here to show the percentage of 6mA sites.

4.       Line 266, ‘GES’ should be explained as ‘GSS”.

5.       Line 321-324, (I) in the legend could not be found in figure 2 and it was not cited in the manuscript. Please delete it.

6.       Line 451-454, the reference format.

Author Response

Thank you for your professional suggestions. We revised the format of citing references to make them consistent and compliant. The responses to your minor concerns are as follows:

1. Line 45, What is ‘ONT data’? please explain.

Response: The ‘ONT data’ was sequencing data from Oxford Nanopore Technologies, we added the full name of ONT in line 45:

The development of methodologies for detecting modified bases from ONT (Oxford Nanopore Technologies) data offers an avenue for 6mA identification.

2. Line 45-46, the format of the references in the parenthesis is not consistent with others. Please keep them consistent. Besides, these references could not be found in the Reference section.

Response: We revised the format of the reference and added the references in the Reference section.

3. Line 226, the Supplemental Fig. S4 is the consensus motif, which could not be cited here to show the percentage of 6mA sites.

Response: Thank you for pointing out this error. We deleted the “Supplemental Fig. S4” in line 223.

4. Line 266, ‘GES’ should be explained as ‘GSS”.

Response: The ‘GES’ was ‘gene end site’, we added the full name of GES in line 266 to make it more clear.

5. Line 321-324, (I) in the legend could not be found in figure 2 and it was not cited in the manuscript. Please delete it.

Response: We deleted the excrescent (I) in line 317.

6. Line 451-454, the reference format.

Response: We revised the reference format in lines 457-460 and doubled check the reference format in the manuscript to make them consistent and compliant.